# Broad Vitamin B_6_-Related Metabolic Disturbances in a Zebrafish Model of Hypophosphatasia (TNSALP-Deficiency)

**DOI:** 10.3390/ijms26073270

**Published:** 2025-04-01

**Authors:** Jolita Ciapaite, Monique Albersen, Sanne M. C. Savelberg, Marjolein Bosma, Nils W. F. Meijer, Federico Tessadori, Jeroen P. W. Bakkers, Gijs van Haaften, Judith J. Jans, Nanda M. Verhoeven-Duif

**Affiliations:** 1Department of Genetics, University Medical Center Utrecht, 3584 EA Utrecht, The Netherlands; m.albersen@amsterdamumc.nl (M.A.); m.bosma@umcutrecht.nl (M.B.); n.w.f.meijer@umcutrecht.nl (N.W.F.M.); g.vanhaaften@umcutrecht.nl (G.v.H.); j.j.m.jans@umcutrecht.nl (J.J.J.); n.verhoeven@umcutrecht.nl (N.M.V.-D.); 2Hubrecht Institute-KNAW and University Medical Center Utrecht, 3584 CT Utrecht, The Netherlands; f.tessadori@hubrecht.eu (F.T.); j.bakkers@hubrecht.eu (J.P.W.B.); 3Department of Medical Physiology, University Medical Center Utrecht, 3584 CM Utrecht, The Netherlands

**Keywords:** vitamin B_6_, hypophosphatasia, alpl deficiency, zebrafish, direct-infusion high-resolution mass spectrometry

## Abstract

Hypophosphatasia (HPP) is a rare inborn error of metabolism caused by pathogenic variants in *ALPL*, coding for tissue non-specific alkaline phosphatase. HPP patients suffer from impaired bone mineralization, and in severe cases from vitamin B_6_-responsive seizures. To study HPP, we generated *alpl*^-/-^ zebrafish using CRISPR/Cas9 gene-editing technology. At 5 days post fertilization (dpf), no *alpl* mRNA and 89% lower total alkaline phosphatase activity was detected in *alpl*^-/-^ compared to *alpl*^+/+^ embryos. The survival of *alpl*^-/-^ zebrafish was strongly decreased. Alizarin red staining showed decreased bone mineralization in *alpl*^-/-^ embryos. B_6_ vitamer analysis revealed depletion of pyridoxal and its degradation product 4-pyridoxic acid in *alpl*^-/-^ embryos. Accumulation of d3-pyridoxal 5′-phosphate (d3-PLP) and reduced formation of d3-pyridoxal in *alpl*^-/-^ embryos incubated with d3-PLP confirmed Alpl involvement in vitamin B_6_ metabolism. Locomotion analysis showed pyridoxine treatment-responsive spontaneous seizures in *alpl*^-/-^ embryos. Metabolic profiling of *alpl*^-/-^ larvae using direct-infusion high-resolution mass spectrometry showed abnormalities in polyamine and neurotransmitter metabolism, suggesting dysfunction of vitamin B_6_-dependent enzymes. Accumulation of N-methylethanolaminium phosphate indicated abnormalities in phosphoethanolamine metabolism. Taken together, we generated the first zebrafish model of HPP that shows multiple features of human disease and which is suitable for the study of the pathophysiology of HPP and for the testing of novel treatments.

## 1. Introduction

Hypophosphatasia (HPP) is a rare inborn error of metabolism that affects the development of bones and teeth and is caused by pathogenic variants in the *ALPL* gene, coding for the tissue-nonspecific isozyme alkaline phosphatase (TNSALP, EC:3.1.3.1). Since the first description by Rathbun [1], more than 400 variants in *ALPL*, predominantly missense, have been identified [2], explaining the highly variable clinical phenotype of HPP. Both autosomal recessive and autosomal dominant modes of inheritance have been reported, with the former often, but not always, corresponding to a more severe clinical phenotype [3]. TNSALP is expressed in the liver, bone (synthesized by the osteoblasts), and kidney, as well as the brain [4], and it functions as a homodimeric ectoenzyme with a broad phospho-substrate specificity [5]. Natural substrates of TNSALP include, but likely are not limited to [5], inorganic pyrophosphate (PP_i_) [6,7], pyridoxal 5′-phosphate (PLP, the active form of vitamin B_6_) [8,9,10], and phosphoethanolamine [8,11,12]. The severity of clinical phenotype strongly correlates with residual TNSALP enzyme activity [3,5,13]. Increased levels of plasma PLP and urinary and plasma PEA along with reduced serum unfractionated alkaline phosphatase activity serve as diagnostic markers of HPP [5,14].

Accumulation of TNSALP substrates in HPP patients reflects the physiological role of the enzyme and clarifies the metabolic basis of HPP. Based on the age of diagnosis/onset of symptoms and severity of clinical phenotype, HPP is classified as perinatal (benign), perinatal (severe), infantile, childhood (mild), childhood (severe), adult, or odontohypophosphatasia, with the perinatal (severe) and infantile forms being most severe [14]. The main clinical feature of HPP is abnormal bone mineralization causing premature loss of deciduous teeth, rickets in children, and osteomalacia in adults. The bone phenotype of HPP is explained by accumulation of PP_i_, a potent inhibitor of bone mineralization [15]. Paradoxically, in some cases, premature closure of cranial sutures (craniosynostosis) occurs in infantile and childhood HPP, causing intracranial hypertension [5,16]. In addition, blocked entry of minerals into the skeleton may lead to hypercalcemia/hypercalciuria, nephrocalcinosis, and renal impairment [5,17]. In the most severe forms of perinatal and infantile HPP, vitamin B_6_-dependent seizures may occur, indicating a lethal prognosis [18,19,20,21,22]. The seizures begin in the first hours after birth and are refractory to standard anticonvulsant drugs, but are responsive to pyridoxine (unphosphorylated form of vitamin B_6_) [18,19,20,21,22]. B_6_-dependent seizures are explained by the role of TNSALP in the cellular uptake of PLP (phosphorylated form of vitamin B_6_) [23], with the decreased TNSALP activity presumably leading to vitamin B_6_ deficiency in the central nervous system (CNS) (Figure 1a). The correlation between the response to pyridoxine and the severity of pediatric HPP reinforces TNSALP’s role in vitamin B_6_ metabolism [24]. Other, less well understood neurological symptoms of HPP may include depression, memory loss, ADHD, anxiety, headache, and sleep disturbance [25].

HPP is an incurable disease. In addition to symptomatic treatment, enzyme replacement therapy with asfotase alfa, a mineral-targeted human recombinant TNSALP, is available for treatment of the bone phenotype of HPP [5,26]. Therefore, understanding TNSALP function in the kidney, liver, brain, and other soft tissues as well as the mechanistic basis of milder neurological symptoms of HPP is becoming more relevant in improving quality of life of HPP patients.

The zebrafish (*Danio rerio*) is a promising model organism in the study of human diseases [27], including HPP [28,29]. Zebrafish have four genes coding for alkaline phosphatases: two for intestinal alkaline phosphatases, *alpi.1* and *alpi.2* (gene duplication), one for alkaline phosphatase 3, *alp3* (also expressed in the intestine)*,* and one for tissue-nonspecific alkaline phosphatase *alpl*, which also shows a high degree of genetic conservation with human *ALPL* [28,29]. In the present study, we generated the first *alpl*^-/-^ zebrafish line using CRISPR/Cas9 gene-editing technology. Biochemical and behavioral characterization of *alpl*^-/-^ zebrafish showed that they display multiple features of infantile HPP.

## 2. Results

### 2.1. Generation of the alpl Knockout Zebrafish

The *alpl* knockout zebrafish were generated using CRISPR/Cas9 gene editing. Sanger sequencing of F1 embryos derived from outcrossing F0 mosaic zebrafish revealed a 10 base pair out-of-frame deletion (c.623_632del, p.Gln180fs), predicted to result in a truncated protein (Figure 1b,c). Sequencing results also showed an intact reference sequence at the CRISPR target site in exon 6. The genotyping results of all untreated 5 dpf old embryos used in the present study (*n* = 1752) showed the following genotype distribution: 27% *alpl*^+/+^, 51% *alpl*^+/-^, and 22% *alpl*^-/-^. Upon visual inspection at 5 dpf, *alpl*^-/-^ embryos were morphologically normal, indistinguishable from *alpl*^+/+^ and *alpl*^+/-^ embryos. Compared to the *alpl*^+/+^ embryos, ~50% (*p* < 0.0001) less *alpl* mRNA was detected in *alpl*^+/-^ and virtually no *alpl* mRNA was detected in *alpl*^-/-^ zebrafish embryos at 5 dpf (Figure 1d). Total alkaline phosphatase enzyme activity measured in the whole embryo extracts was 37% (*p* < 0.0001) and 87% (*p* < 0.0001) lower in *alpl*^+/-^ and *alpl*^-/-^ zebrafish embryos, respectively, compared to WT (Figure 1e). The contribution of other alkaline phosphatase isoforms, such as intestinal alkaline phosphatase encoded by *alpi.1* and *alpi.2*, and possibly alkaline phosphatase 3, which is also expressed in the intestine, could explain the residual enzyme activity measured in *alpl*^-/-^ embryos, since intestinal isoforms are insensitive to (-)-tetramisole HCl [30]. In contrast to WT (*alpl*^+/+^) zebrafish, no Alpl protein was detected at 62.5 kDa (the predicted molecular weight of alpl protein (NP_957301.2)) by Western blotting in *alpl*^-/-^ embryos at 5 dpf (Figure 1f).

### 2.2. Abnormal Bone Mineralization, Vitamin B_6_ Metabolism and Locomotion in alpl^-/-^ Embryos

Decreased bone mineralization due to accumulation of pyrophosphate, one of the substrates of TNSALP, is a key feature of HPP [7,8]). To determine the consequences of *alpl* knockout on bone mineralization, we stained 5 dpf embryos with acid-free alizarin red (bone) and alcian blue (cartilage) double stain. Less alizarin red staining of mineralized structures, such as notochord [31], in *alpl*^-/-^ compared to WT and *alpl*^+/-^ embryos at 5 dpf suggested a negative effect of alpl deficiency on bone mineralization (Figure 2a).

Another important TNSALP substrate is extracellular (circulating) PLP (Figure 1a). We assessed the consequences of *alpl* knockout on PLP metabolism by measuring the utilization of d3-PLP by live 5 dpf old embryos. After 3 h incubation with 100 µM d3-PLP in embryo water, significantly more d3-PLP (Figure 2b) and significantly less d3-PL (Figure 2c) was observed in *alpl*^-/-^ compared to WT and *alpl*^+/-^ embryos, indicating impaired d3-PLP hydrolysis to d3-PL.

The steady-state PLP concentration measured in whole embryo extracts was slightly but significantly lower in *alpl*^-/-^ compared to WT and *alpl*^+/-^ embryos (Figure 2d). Since Alpl deficiency is expected to result in increased PLP concentration in fluid extracellular matrix and decreased intracellular PLP concentration (see Figure 1a), the lower PLP concentration measured in whole *alpl*^-/-^ embryo extracts likely indicates that cell-derived PLP dominated the total measured PLP. Concentrations of PL and its degradation product 4-PA were strongly decreased in *alpl*^-/-^ compared to WT and *alpl*^+/-^ embryos (Figure 2e and Figure 2f, respectively), in agreement with the predicted negative effect of Alpl deficiency on both the extra- and intracellular concentrations of these compounds. PMP concentration was not significantly changed in *alpl*^-/-^ compared to WT and *alpl*^+/-^ embryos (Figure 2g). Negligible concentrations of PM and PNP were measured in whole embryo extracts, which were similar in all three genotypes (Figure 2h and Figure 2i, respectively).

Intracellular PLP deficiency can have negative effects on neurotransmitter biosynthesis, leading to vitamin B_6_-dependent seizures [28]. In zebrafish embryos, seizures cause typical fast, circular swimming bursts [32,33,34,35]. Therefore, we recorded the locomotion of 5 dpf old WT, *alpl*^+/-^, and *alpl*^-/-^ embryos with a Zebrabox (Figure 3a and Appendix A). Locomotion analysis revealed that inactivity count (i.e., number of times spent below the inactivity threshold) was significantly lower in *alpl*^-/-^ compared to WT embryos (Figure 3b) without a change in the duration of inactivity (Figure 3c). Furthermore, the count, duration, and distance swum in small movements (speed < 30 mm/s) was significantly lower in *alpl*^-/-^ compared to WT embryos (Figure 3d and Figure 3e, respectively). In contrast, the count, duration, and distance swum in burst movements (speed > 30 mm/s) was significantly increased in *alpl*^-/-^ compared to WT embryos (Figure 3g, Figure 3h and Figure 3i, respectively), indicating spontaneous seizures in the former. No locomotion parameters of *alpl*^+/-^ embryos were significantly different from WT (Figure 3).

### 2.3. Pyridoxine Treatment Normalizes Locomotion and Some Metabolic Abnormalities in alpl^-/-^ Zebrafish Embryos

Next, we investigated how vitamin B_6_ treatment affects locomotion and biochemical abnormalities in 5 dpf old *alpl*^-/-^ zebrafish embryos. Initial experiments showed no significant effects of a single, 3 h treatment with 100 µM pyridoxine (PN) on B_6_ vitamers and locomotion parameters in WT and *alpl*^-/-^ embryos. Increasing the duration of the treatment with 100 µM PN to 72 h resulted in a strong increase in PN concentrations of a similar magnitude in WT and *alpl*^-/-^ embryos (Figure 4a). PN treatment had no significant effect on the PLP concentration in WT embryos (Figure 4b). In contrast, in *alpl*^-/-^ embryos, PN treatment led to an increase in PLP concentration to a level that was significantly higher than in WT (Figure 4b). Concentrations of PL and its degradation product 4-PA increased in both WT and *alpl*^-/-^ embryos in response to PN treatment (Figure 4c and Figure 4d, respectively). However, accumulation of both PL and 4-PA was less pronounced in *alpl*^-/-^ embryos (Figure 4c and Figure 4d, respectively). Concentrations of PM, PMP, and PNP were not significantly affected by PN treatment (Appendix A, respectively). Locomotion analysis showed that 72 h of PN treatment resulted in a significant reduction in parameters indicating spontaneous seizures in *alpl*^-/-^ embryos. Specifically, burst activity count (Figure 4e), duration of burst activity (Figure 4f), and distance swum during burst activity (Figure 4g) were significantly lower in PN-treated compared to untreated *alpl*^-/-^ embryos, and were not significantly different from WT. Furthermore, the concentration of the inhibitory neurotransmitter γ-aminobutyric acid (GABA), which was significantly lower in untreated *alpl*^-/-^ embryos, was restored after PN treatment to the WT level (Figure 4h). Since PLP is a cofactor in many enzymes involved in amino acid metabolism, we analyzed amino acids. Glutamate, glutamine, and asparagine concentrations were significantly lower in untreated *alpl*^-/-^ compared to WT embryos. These amino acids were normalized after 72 h of PN treatment (Figure 4i and Appendix A). Methionine, the only amino acid that significantly accumulated in untreated *alpl*^-/-^ embryos, was normalized to WT level after PN treatment (Figure 4i and Appendix A). Furthermore, concentrations of several amino acids significantly increased, independent of genotype, in response to PN treatment (proline, alanine, valine, leucine, serine, tryptophan, and histidine), and the concentration of only ornithine decreased, independent of genotype, in response to PN treatment (Appendix A). The concentrations of the remaining quantified amino acids were not affected either by genotype nor PN treatment (Appendix A).

To explore the global metabolic response to PN treatment we analyzed the metabolomes of untreated and 100 μM PN-treated WT and *alpl*^-/-^ embryo extracts by direct-infusion high-resolution mass spectrometry (DI-HRMS). Using an in-house developed peak calling pipeline [36], 1919 mass peaks were annotated with 3941 metabolites that could occur endogenously. Partial least squares-discriminant analysis (PLS-DA) showed a clear discrimination of *alpl*^-/-^ and WT metabolomes, which became less pronounced upon PN treatment, suggesting normalization of *alpl*^-/-^ zebrafish metabolome after 72 h treatment with 100 µM PN (Figure 5a). Among the metabolites of component 1, identified by PLS-DA as contributing the most to the separation of the data, were several phosphorylated compounds (adenosine monophosphate (AMP), glycerol 3-phosphate, uridine 5′-monophosphate (UMP)), the vitamin B_6_ degradation product 4-pyridoxic acid (4-PA), as well as several amino acids and their derivatives (glutamine, GABA, methionine, methionine sulfoxide, and cystathionine) (Figure 5b). Statistical analysis of the data using one-way ANOVA with Tukey’s post-hoc test identified 284 significantly altered metabolites (Appendix A). The heatmap overview of the 50 highest-ranking metabolites based on one-way ANOVA results is shown in Figure 5c. Closer inspection of Tukey’s post-hoc test results showed that 144 metabolites identified by one-way ANOVA were significantly changed in *alpl*^-/-^ embryos compared to WT, of which 64 metabolites were subsequently significantly affected by PN treatment (Appendix A). Among phospho compounds that were increased in *alpl*^-/-^ embryos compared to WT, but did not respond to PN treatment and therefore could be directly related to the alpl phosphatase function, were: inositol cyclic phosphate, cytidine monophosphate, phosphoguanidinoacetate, (S)-5-diphosphomevalonic acid, phosphodimethylethanolamine, and glycerylphosphorylethanolamine (Appendix A). In contrast, the level of O-phosphoethanolamine was normalized by PN treatment, indicating that its accumulation was caused by a different, vitamin B_6_-dependent, mechanism (Appendix A). Figure 5d shows a selection of highest-ranking metabolites that were responsive to PN treatment. The localization of these metabolites within the metabolic pathways is marked based on the KEGG pathway database information in Figure 5e. Abnormalities in methionine, its oxidation product methionine sulfoxide, and cystathionine levels in *alpl*^-/-^ embryos (Figure 5d) suggested impaired functioning of methionine and/or folate cycles and the transsulfuration pathway, which in turn is important for synthesis of glutathione (Figure 5e). These pathways contain vitamin B_6_-dependent enzymes (such as CBS and CGL; see Figure 5e), and the fact that all three metabolites were normalized after PN-treatment indicated that the effects were likely mediated by these enzymes. While glutathione levels were similar in WT and *alpl*^-/-^ embryos, they significantly increased in both genotypes upon PN treatment (Appendix A), indicating the sensitivity of glutathione synthesis to vitamin B_6_ availability. Increased levels of 2-hydroxybutyric acid and propionylcarnitine in *alpl*^-/-^ embryos suggested that surplus cystathionine was diverted from the transsulfuration pathway, possibly due to impaired activity of vitamin B_6_-dependent cystathionine-β-synthase (Figure 5d,e). Moreover, the impairment of transsulfuration pathway activity could underlie the elevation of N1-acetylspermidine levels in *alpl*^-/-^ embryos (Appendix A) by pushing the methionine cycle intermediates into the polyamine pathway (Figure 5e). While none of the measured canonic folate cycle intermediates were altered in *alpl*^-/-^ embryos, we observed accumulation of 10-formyldihydrofolate (Appendix A), which has been shown to participate in purine synthesis in mammalian cells [37]. Consequently, we found increased levels of AMP and inosine in *alpl*^-/-^ embryos, which decreased after PN treatment (Figure 5d,e). Additionally, PN treatment-responsive abnormalities in the pyrimidine synthesis pathway intermediates UMP (Figure 5d,e) and orotidine (Appendix A) in *alpl*^-/-^ embryos may have resulted from abnormal folate cycle activity (Figure 5e).

### 2.4. Progression of Metabolic Abnormalities and Decreased Survival in alpl^-/-^ Zebrafish Larvae

To assess the progression of metabolic abnormalities, we analyzed 10 dpf old WT and *alpl*^-/-^ zebrafish larvae. Only 24% of *alpl*^-/-^ larvae were alive at 10 dpf (Figure 6a). Treatment with 100 μM PN for 5 days (from 5 dpf to 10 dpf, 30 min/day) led to a significant improvement in the survival of *alpl*^-/-^ zebrafish larvae (Figure 6a); however, it did not restore the survival to WT levels. While this treatment regimen resulted in increases in PN and PLP concentrations in *alpl*^-/-^ zebrafish larvae, the concentrations of PL and its degradation product 4-PA remained low (Figure 6b). Locomotion analysis using Zebrabox did not indicate spontaneous seizures during 1 h of measurement in the surviving larvae, as indicated by similar burst activity parameters in WT and *alpl*^-/-^ larvae (Figure 6c and Appendix A). However, there was a decrease in all small activity parameters, i.e., distance swum in small movements (Figure 6c), as well as the count and duration of small movements in *alpl*^-/-^ larvae (Appendix A), indicating decreased mobility of the mutants, which was normalized by PN treatment. No differences were observed in the inactivity count and duration (Appendix A). The concentration of GABA was lower in untreated *alpl*^-/-^ larvae, but it was normalized by PN treatment (Figure 6d).

Next, we analyzed the metabolomes of 10 dpf old WT and *alpl*^-/-^ larvae with DI-HRMS. Consequently, 1899 mass peaks were annotated with 3891 metabolites that could occur endogenously [36]. Statistical analysis using a *t*-test identified 183 metabolites significantly (*p* < 0.05) altered in *alpl*^-/-^ compared to WT larvae (Appendix A). Further analysis using a volcano plot identified several metabolites that were strongly and significantly changed in *alpl*^-/-^ larvae (Figure 6e, Appendix A). Statistical analysis revealed abnormalities in several neurotransmitters, including decreased levels of GABA (Appendix A) and increased levels of L-DOPA in *alpl*^-/-^ larvae (Figure 6f, Appendix A). Accumulation of L-DOPA in conjunction with the elevation of vanillactic acid (Figure 6f) pointed towards abnormal activity of the PLP-dependent aromatic L-amino acid decarboxylase (AADC, EC: 4.1.1.28) [38]. Consequently, the downstream products of AADC activity, dopamine and epinephrine, were negatively affected by Alpl deficiency (Figure 6f). 2-(3-Carboxy-3-(methylammonio)propyl)-L-histidine, a post-translationally modified histidine that serves as a substrate for diphthine synthase (EC:2.1.1.98), a methyltransferase involved in transfer of one-carbon groups, was strongly decreased in *alpl*^-/-^ larvae (Figure 6e, Appendix A). Furthermore, changed levels of several polyamine species (N-acetylcadaverine, N-acetylputrescine, dehydrospermidine, and N1-acetylspermidine (all decreased) and norspermidine (increased)) in *alpl*^-/-^ zebrafish larvae (Figure 6e, Appendix A) indicated altered activity of a polyamine synthesis pathway. An increase in vitamin A (retinol) and decrease in retinal (oxidized form of retinol, component of visual pigment [39]) (Figure 6g, Appendix A) pointed towards abnormalities in vitamin A metabolism in *alpl*^-/-^ larvae. Moreover, accumulation of N-methylethanolaminium phosphate in *alpl*^-/-^ larvae (Figure 6h) suggested abnormal metabolism of phosphoethanolamines.

## 3. Discussion

While the mechanistic basis of the bone phenotype of HPP is well understood, several unexplained clinical features remain, including craniosynostosis and neurological manifestations [22,25]. Moreover, the understanding of TNSALP function in the soft tissues, such as the kidney, liver, and brain, is very limited, underscoring the need for further fundamental research. In the present study we described the first genetic model of tissue non-specific alkaline phosphatase deficiency in zebrafish that displayed many key features of human HPP. Our data showed that like TNSALP in humans, Alpl in zebrafish has a function in vitamin B_6_ metabolism, as illustrated by strongly impaired ability of *alpl*^-/-^ embryos to hydrolyze d3-PLP to d3-PL. The deficiency in Alpl activity led to lower total PLP, PL, and 4-PA levels, and to vitamin B_6_ (pyridoxine)-responsive seizures. Moreover, multiple metabolic abnormalities were identified through untargeted metabolomics that were linked to decreased cellular/tissue PLP levels and impaired activity of PLP-dependent enzymes. However, pyridoxine treatment improved, but did not fully restore to WT levels, the survival of *alpl*^-/-^ zebrafish. This suggests that a deficiency of functions other than those involved in vitamin B_6_ metabolism (e.g., bone mineralization) plays an essential role in the lethality of Alpl deficiency in zebrafish.

There is a high degree of evolutionary conservation of the gene coding for TNSALP in vertebrates [40,41], indicating that animal models of TNSALP deficiency can yield valuable insights into the physiological functions of TNSALP and pathophysiology of HPP [29]. Murine models of TNSALP deficiency have already proven essential for the development of enzyme replacement therapy [42], which is currently the only available treatment option effective for the bone phenotype of human HPP [26]. The zebrafish is increasingly used as an alternative model organism to study human diseases due to its ease of breeding, large number of offspring, and short generation time. The conservation of zebrafish Alpl function in the skeleton and nervous system was previously postulated based on the comparison of tissue-specific gene expression patterns in the zebrafish, mouse, and human [41]. In the present study we describe the first genetic model of TNSALP deficiency in zebrafish generated using CRISPR/Cas gene editing. Although there was virtually no mRNA nor protein detectable, there was residual alkaline phosphatase activity in 5 dpf *alpl*^-/-^ embryos. Since enzyme activity measurements were performed in the total embryo extracts, the most plausible explanation of the residual activity is the contribution of intestinal alkaline phosphatase (encoded by *alpi.1* and *alpi.2*) and alkaline phosphatase 3, also expressed in the intestine, to the total measured enzyme activity. Residual alkaline phosphatase activity was also measured in the serum of the first genetic murine model of TNSALP deficiency (*Akp2^-/-^*), which was explained by the contribution of genetically distinct intestinal alkaline phosphatase activity [43]. Furthermore, the biochemical and behavioral characteristics of *alpl*^-/-^ zebrafish described in the present study were comparable to the phenotypic features of *Akp2^-/-^* mice, which recapitulate lethal infantile HPP extremely well, including bone abnormalities and vitamin B_6_-responsive seizures, with untreated seizing animals dying before weaning [43,44,45]. Data from available murine models of TNSALP deficiency show that in contrast to human TNSALP, murine TNSALP appears to not be essential in the initial events of bone mineralization during intrauterine development (i.e., no severe skeletal abnormalities typical to perinatal HPP), but it becomes important for this process after birth [43,44,45]. In the present study, we showed that zebrafish Alpl functions in bone mineralization, as indicated by decreased alizarin staining of notochord, one of the earliest mineralizing structures in zebrafish embryos [31]. This effect was comparable to the effect of chemical Alpl inhibition on bone mineralization in wild-type zebrafish embryos [41].

Vitamin B_6_ (pyridoxine)-responsive seizures are a rare clinical feature of HPP, observed only in the most severe forms of perinatal and infantile HPP [18,19,20,21,22]. They are presumably caused by decreased PLP and PL availability in the cells of the central nervous system due to decreased/absent hydrolysis of extracellular PLP to PL (Figure 1a). Impaired activity of PLP-dependent enzymes involved in amino acid and neurotransmitter synthesis, e.g., decreased synthesis of the inhibitory neurotransmitter GABA due to lower activity of PLP-dependent glutamate decarboxylase (EC: 4.1.1.15), and a resulting imbalance in the levels of inhibitory and excitatory neurotransmitters, underlies seizures [18,46,47]. Indeed, by using stably labeled PLP, we could show that the hydrolysis of PLP and production of PL is strongly impaired in *alpl*^-/-^ embryos, leading to lower concentrations of PLP and PL, as well as 4-PA (degradation product of PL) in total embryo extracts (dominated by tissue derived PLP and PL). Unfortunately, due to the small size of the embryos, we were unable to separately quantify PLP in the circulation and in the individual tissues. However, PLP deficiency in tissues including the brain were suggested by the observation that GABA concentrations in *alpl*^-/-^ 5 dpf embryos and 10 dpf larvae were lower than in WT zebrafish, and they were normalized by PN treatment. The abnormalities in B_6_ vitamer and GABA concentrations in *alpl*^-/-^ zebrafish were in line with the findings in *Akp2^-/-^* mice, which have high serum PLP and low PL concentrations, as well as low PLP and PL concentrations in various tissues including the brain [43,44], and low brain GABA concentrations [43]. In HPP patients, circulating PLP concentration is elevated [9], making it a good biomarker of HPP [14], while PL concentration is less often reported, ranging from normal [18] to low in severe cases [10]. A single report in various post-mortem tissues of two patients with perinatal HPP showed no alterations in PLP and PL concentrations [10]. Lastly, similar to *Akp2^-/-^* mice [43,44,45] and severe forms of perinatal and infantile HPP [18,19,20,21,22], *alpl*^-/-^ zebrafish embryos also developed spontaneous seizures that were responsive to PN treatment. However, we detected no spontaneous seizures in 10 dpf *alpl*^-/-^ larvae, which may possibly be attributed to the fact that the measurements in larvae were conducted in a small number of the surviving larvae. Only a subpopulation of *Akp2^-/-^* mice experience seizures [43]. The underlying cause of phenotypic heterogeneity despite genetic homogeneity is not clear. We showed that, similarly to *Akp2^-/-^* mice [43,44,45] and HPP patients [18,19,20,21,22], spontaneous seizures in *alpl*^-/-^ zebrafish embryos were responsive to PN treatment, leading to improved survival of the mutants. The lack of complete rescue of survival by PN treatment is in line with the multiple non-overlapping TNSALP functions, as demonstrated in *Akp2^-/-^* mice, where PN treatment prevents seizures without a beneficial effect on the skeletal phenotype [48].

The untargeted metabolomics analysis of the broader consequences of Alpl deficiency showed abnormalities in several neurotransmitter levels attributable to decreased cellular PLP availability. Next to lower GABA levels, increased levels of L-DOPA and vanillactic acid along with lower levels of dopamine and epinephrine pointed towards decreased activity of PLP-dependent AADC in *alpl*^-/-^ larvae. Impaired AADC activity was also implied in HPP patients based on elevated 3-ortho-methyldopa in CSF [18,47] and increased vanillactic acid in urine [47]. Moreover, we observed that accumulation of N-methylethanolaminium phosphate (PEA) is detectable in *alpl*^-/-^ larvae, but not yet in 5 dpf embryos, indicating that in zebrafish accumulation of PEA develops gradually. In *Akp2^-/-^* mice, elevated serum PEA concentrations were shown in 8–10 day old pups; however, age dependence was not investigated [43]. Furthermore, we found increased methionine and cystathionine levels in *alpl*^-/-^ embryos, suggesting altered activity of the methionine cycle and transsulfuration pathway, likely caused by the impaired activity of PLP-dependent enzymes, as underscored by the normalization of these metabolites in response to PN treatment (Figure 5e). Interestingly, an elevation in methionine and cystathionine levels was also reported in the brains of 1-week-old *Akp2^-/-^* mice [49], suggesting common underlying mechanisms in zebrafish and mice. Furthermore, elevated AMP and inosine levels in *alpl*^-/-^ embryos suggest abnormalities in purine metabolism that could be linked to the deficiency in Alpl ectophosphatase function and to the mechanisms of chronic pain via an effect on circulating adenosine levels [50]. Our observation that PN treatment led to normalization of AMP and inosine levels in *alpl*^-/-^ embryos suggests that changes in these compounds were caused by reduced vitamin B_6_ availability rather than by the ectophosphatase activity of Alpl. However, empirical pyridoxine therapy for chronic fatigue and pain in four adult-onset HPP patients did not provide symptomatic relief [22], suggesting that the underlying mechanisms are vitamin B_6_-independent. It must be noted that due to the contribution of isobaric compounds to the levels of AMP, inosine, and adenosine (not changed in *alpl*^-/-^ embryos determined with DI-HRMS, future follow up research using targeted methods is required to clarify the involvement of Alpl in the regulation of purine metabolism. This could contribute to better understanding of the mechanisms of chronic pain and help to develop new treatments to improve the quality of life of HPP patients.

Lastly, we observed accumulation of vitamin A (retinol) and decreased levels of retinal in *alpl*^-/-^ larvae. Interestingly, high *alpl* expression and alpl enzyme activity were observed in the eyes (especially lens and retina) of zebrafish embryos [41], as well as retina of other vertebrates [51], suggesting that TNSALP has a function in vision. However, no eye-specific phenotype was reported in HPP patients nor TNSALP deficient mice. Abnormalities in vitamin A metabolism have been implicated in the development of craniosynostosis and skeletal abnormalities in humas and zebrafish [52]. The mechanistic basis of how Alpl deficiency leads to retinol accumulation needs further investigation. Possibly, impaired Ca^2+^ homeostasis caused by Alpl deficiency could affect retinol transport into the cell, which is regulated by Ca^2+^/calmodulin [53], leading to accumulation of circulating retinol and decreased intracellular retinal production.

## 4. Materials and Methods

### 4.1. Zebrafish Maintenance and Treatment Protocols

Zebrafish (*Danio rerio*) were raised and maintained under standard laboratory conditions [31]. Animal experiments were approved by and performed according to the guidelines of the Animal Welfare Body Utrecht, Utrecht University (protocol code 1444WP2B2).

Pyridoxine-treatment experiments were carried out in either 5 dpf embryos and 10 dpf larvae. For embryo treatment, batches of 2 dpf embryos (not genotyped), generated by incrossing *alpl*^+/-^ parents, were randomly assigned to untreated or pyridoxine-treated groups. Next, 0 µM (untreated) or 100 µM (treated) pyridoxine (Sigma-Aldrich, Zwijndrecht, The Netherlands) was added to E3 medium in a petri dish and embryos were raised to 5 dpf (3 days continuous treatment). Media were refreshed every 24 h. At 5 dpf, embryos were anesthetized with tricaine, a sample of caudal fin was dissected for DNA isolation and genotyping, and embryos were instantly frozen on dry ice and stored at −80 °C until analysis. For larvae treatment, zebrafish embryos were genotyped at 3 dpf as described in [34]. Only *alpl*^+/+^ and *alpl*^-/-^ embryos were raised to 5 dpf. Starting from 5 dpf, zebrafish were treated for 5 consecutive days (between 9:00 and 10:30 am) with 0 µM or 100 µM pyridoxine for 30 min (Appendix A). Treatment included placing zebrafish larvae (*n* = 18–22 per tank) in plastic tanks containing 500 mL of system water without pyridoxine (untreated) or 500 mL of system water containing 100 µM pyridoxine (treated). After 30 min, zebrafish larvae were rinsed and placed in the home tank. At 10 dpf, zebrafish larvae were anesthetized with tricaine and terminated by instant freezing on dry ice (1 larva per Eppendorf cup) and stored at −80 °C until analysis.

To assess the utilization of stably labeled pyridoxal 5′-phospate, 5 dpf old embryos (~60 embryos/petri dish) were incubated with 100 µM pyridoxal-5′-phosphate (methyl-D3) (d3-PLP) (Buchem, Minden, The Netherlands) in E3 for 0, 1, 2, and 3 h. At specified time-points, embryos were washed with E3, anesthetized with tricaine, dissected for genotyping, snap-frozen on dry ice and stored at −80 °C until analysis.

### 4.2. sgRNA and Cas9 mRNA Design and Synthesis

CRISPR/Cas9 gene-specific regions for *alpl* were designed by the Sanger Institute (Hinxton, Cambridge, UK) using a modified version of CHOPCHOP (http://chopchop.cbu.uib.no (accessed on 11 January 2015)). Target sites were selected in exon 5 and exon 6 (Appendix A) [54,55]. The gene-specific oligonucleotides contained the T7 promotor sequence (5′-TAATACGACTCACTATA-3′), the GGN20 target site without the Protospacer Adjacent Motif (PAM), and the constant complementary region 5′-GTTTTAGAGCTAGAAATAGCAAG-3′. Oligonucleotides were ordered at IDT (Integrated DNA Technologies, Coralville, IA, USA) and the zebrafish specific pCS2-nCas9n plasmid was obtained from Addgene (Cambridge, MA, USA). Cas9 mRNA transcription and sgRNA synthesis were performed as described before [56].

### 4.3. Generation of alpl Knockout Zebrafish

Wild type Tupfel longfin (WT TL) one-cell stage zebrafish embryos were microinjected in the yolk with approximately 1 nl sgRNA mixture (sgRNA targeting exon 5 and 6, each 30 ng/µL) and Cas9 mRNA (250 ng/µL). CRISPR efficiency was determined in a subpopulation of healthy microinjected larvae at 4 dpf. The rest of the healthy microinjected larvae were raised till adulthood. Heterozygous variation was assessed in DNA extracted from healthy embryonal offspring (F1) at 24 dpf. Offspring from a mosaic founder that contained a 10 bp out-of-frame deletion was raised till adulthood and was fin-clipped for genotyping at 9 weeks of age. The mutant zebrafish line was maintained in the heterozygous form by crossing *alpl*^+/-^ zebrafish with WT TL. In this study, F6 zebrafish (*alpl*^+/+^, *alpl*^+/-^ and *alpl*^-/-^) were used, obtained from incrossing F5 *alpl*^+/-^ zebrafish.

### 4.4. DNA Extraction and Genotyping

Depending on the type of experiment, genotyping was performed on the caudal fin dissections at 3 or 5 dpf (overall experiments), whole 5 dpf embryos (Zebrabox and staining experiments), or adult zebrafish caudal fin dissections (line maintenance) as described in detail in [34]. Briefly, tissue was lysed in single embryo lysis buffer (SEL) containing 10 mM Tris pH 8.2, 10 mM EDTA, 200 mM NaCl, 0.5% sodium dodecyl sulfate (SDS), and 12 U/mL proteinase K (freshly added, Thermo Scientific, Waltham, MA, USA, cat. # EO0491). DNA was isolated using the following thermocycler program: 60 min 60 °C, 15 min 95 °C, 15 min 4 °C, ∞ 12 °C. Genomic regions flanking the CRISPR target sites were amplified with CRISPR site-specific PCR primers (Appendix A), using AmpliTaq Gold 360 DNA polymerase (Applied Biosystems, Waltham, MA, USA, cat. # 4398823) in combination with a touch down PCR program as previously described [57]. Amplicons were visualized on a 3% agarose gel and mutations were confirmed by Sanger sequencing.

### 4.5. RNA Isolation and Real-Time PCR

Zebrafish embryos (5 dpf) were placed in sterile Eppendorf tubes on ice (10 embryos/tube per genotype, 3 tubes/genotype). Sterile, RNAse-free zirconium oxide beads (0.5 mm) and cold 0.5 mL TRI reagent (Sigma-Aldrich, cat. # T9424) were added to each tube. Embryos were homogenized using a bullet blender tissue homogenizer (Next Advance, Troy, NY, USA) for 10 min in stand 8 at 4 °C. Total mRNA was isolated from the embryo homogenates following the manufacturer’s recommendations. The quantity and purity of the total RNA was quantified using a NanoDrop spectrophotometer (Thermo Scientific). One µg of total RNA was reverse transcribed to cDNA using M-MLV reverse transcriptase (Sigma-Aldrich, cat. # M1302) according to the manufacturer’s protocol. Real-time PCR was performed with a StepOne Real-Time PCR System (Applied Biosystems, Waltham, MA, USA) using the SYBR Select Master Mix (Applied Biosystems, cat. # 4472908) and the primers listed in Appendix A. The *alpl* (ZDB-GENE-040420-1, RefSeq:NM_201007.2) mRNA levels were normalized to the mRNA level of β-actin (ZDB-GENE-000329-1, RefSeq:NM_131031.2) and expressed relative to the wild type (calculated according to the ΔΔCt method).

### 4.6. Alizarin Red and Alcian Blue Staining

Mineralized bone and cartilage were stained in whole 5 dpf embryos with acid-free alizarin red and alcian blue double stain as described in [58]. Briefly, 5 dpf embryos were anesthetized with tricaine and up to 20 embryos were collected per 1.5 mL Eppendorf tube. After removing the medium, 1 mL of 4% paraformaldehyde in phosphate buffered saline was added per tube and embryos were fixed for 2 h with agitation at 500 rpm in an Eppendorf thermomixer at room temperature (RT), followed by washing and dehydration with 1 mL 50% ethanol for 10 min at RT. Embryos were stained overnight with 0.0005% alizarin red and 0.4% alcian blue working solution with agitation at RT. Stained embryos were washed and bleached with 1.5% H_2_O_2_ containing 1% KOH for 20 min at RT. After removing the bleach solution, 1 mL 20% glycerol containing 0.25% KOH was added and embryos were incubated for 2 h, followed by overnight incubation with 1 mL 50% glycerol containing 0.25% KOH at RT. Next, medium was replaced with 50% glycerol containing 0.1% KOH and embryos were stored at 4 °C. Images were captured with a Leica DFC420C digital microscope camera (Leica Microsystems, Wetzlar, Germany) mounted on a Zeiss Axioplan brightfield microscope (Carl Zeiss AG, Oberkochen, Germany). After the imaging, DNA was extracted from stained embryos and genotype was determined as described in Section 2.4.

### 4.7. Alkaline Phosphatase Enzyme Activity

Total alkaline phosphatase enzyme activity was determined in the whole embryo homogenates using assay described in [59]. Briefly, 5 dpf embryos (*n* = 30 per genotype) were homogenized in 200 µL of Dulbecco′s Phosphate Buffered Saline (DPBS, Sigma-Aldrich, cat. # D8537) containing 0.1% Triton X100 using a bullet blender tissue homogenizer (Next Advance) for 5 min in stand 8 at 4 °C. Homogenates were centrifuged at 600× *g* for 5 min at 4 °C and sonicated using an ultrasonic disintegrator (Soniprep 150 Plus, MSE, Cholet, France) for 30 s in the pulse mode (1 s on 1 s off, amplitude 10 µm) on ice. The assay mix contained 20 μL of embryo extract (5× diluted in DPBS, final protein concentration in the assay 0.065 mg/mL), 80 μL DPBS, and 100 μL of CSPD ready-to-use reagent (0.25 mM solution; Roche GmbH, Mannheim, Germany, cat. # CSPD-RO) without Alpl inhibitor (-)-tetramisole HCl (Sigma-Aldrich, cat. # L9756) or with 20 mM (-)-tetramisole HCl. Alpl activity was measured by following the chemiluminescence for 5 min at 37 °C using Clariostar microplate reader (BMG Labtech, Ortenberg, Germany). Alpl activity was expressed as RLU/min/mg protein. Protein concentration in the embryo homogenates was determined using a Pierce BCA protein assay kit according to the manufacturer’s protocol (Thermo Scientific, cat. # 23225).

### 4.8. Western Blotting

Zebrafish embryos (5 dpf, *n* = 32 per genotype) were placed in Eppendorf tubes on ice. Then, 150 µL RIPA lysis and extraction buffer (Thermo Scientific, cat. #89900) containing 2 mM NaF (Sigma-Aldrich) and protease inhibitor cocktail (1:200, Roche) was added, followed by zirconium oxide beads (0.5 mm). Embryos were homogenized using a bullet blender tissue homogenizer (Next Advance) for 10 min in stand 8 at 4 °C. Tissue homogenates were solubilized with agitation for 2 h at 4 °C, followed by centrifugation at 16,200× *g* for 10 min at 4 °C. Supernatants were mixed with LDS sample buffer (NuPage, Invitrogen, Waltham, MA, USA, cat. # NP0007) and dithiothreitol (final concentration 50 mM), and denatured at 98 °C for 5 min with agitation. Proteins were resolved on NuPAGE 4–12% Bis-Tris gels (Invitrogen) and transferred to polyvinylidene difluoride (PVDF) membranes (Immobilon-P) with a semi-dry blotting system (Novex, Invitrogen), following the manufacturer’s recommendations. Membranes were blocked with tris-buffered saline (TBS) containing 0.1% Tween 20 (TBS-T) and 50 g/L bovine serum albumin (BSA, Sigma-Aldrich) for 1 h at RT. Next, the membranes were incubated overnight at 4 °C with primary rabbit polyclonal anti-ALPL antibody (1:1000, Sigma-Aldrich, cat. # HPA008765) or mouse monoclonal anti-glyceraldehyde-3-phosphate dehydrogenase (GAPDH, 1:5000, Santa Cruz Biotechnology, Dallas, TX, USA, cat. # sc-365062) in TBS-T containing 10 g/L BSA. After washing three times for 10 min each with TBS-T, membranes were incubated with a corresponding horseradish peroxidase-conjugated secondary antibody in TBS-T containing 5 g/L BSA for 1 h at RT. After the final wash of 3 × 10 min with TBS-T, the immunocomplexes were detected using SuperSignal™ West Atto Ultimate Sensitivity Substrate (Thermo Scientific, cat. # A38554) and images were captured with the ChemiDoc MP imaging system (Bio-Rad Laboratories, Hercules, CA, USA).

### 4.9. B_6_ Vitamer Analysis

Frozen zebrafish 5 dpf embryos (3 embryos/100 µL TCA) or 10 dpf larvae (1 larva/100 µL TCA) were homogenized in ice-cold trichloroacetic acid (TCA; 50 g/L) with zirconium oxide beads (0.5 mm) using a bullet blender tissue homogenizer (Next Advance) at a speed of 8 for 10 min at 4 °C. Homogenates were centrifuged at 16,200× *g* for 5 min at 4 °C. Next, 80 µL of the supernatant was mixed with 80 µL of a solution containing stable isotope-labeled internal standards, vortexed, incubated for 15 min in the dark and centrifuged at 16,200× *g* for 5 min at 4 °C. B_6_ vitamers were quantified using ultra-performance liquid chromatography tandem mass spectrometry (UPLC-MS/MS) as previously described [60], except for using 10 times lower concentrations of the calibration samples. For the analysis of pyridoxal 5′-phosphate-(methyl-d3) (d3-PLP) utilization and pyridoxal-(methyl-d3) (d3-PL) formation, zebrafish embryos were processed and analyzed using the same protocol, except that no stable isotope-labeled internal standards were added during UPLC-MS/MS measurement. During all steps, samples were protected from light as much as possible.

### 4.10. Non-Quantitative Direct-Infusion High-Resolution Mass Spectrometry (DI-HRMS)

Metabolite profiling was performed in 5 dpf embryos and 10 dpf larvae using a non-quantitative DI-HRMS method described in [36]. For extraction of metabolites, three embryos or a single 10 dpf larvae were homogenized in 100 µL of ice-cold 100% methanol with zirconium oxide beads (0.5 mm) using a bullet blender tissue homogenizer (Next Advance Inc., Averill Park, NY, USA) at a speed of 8 for 10 min at 4 °C. Homogenates were centrifuged at 16,200× *g* for 5 min at 4 °C. The supernatants (70 µL) were mixed with 60 µL of 0.3% formic acid (Emsure, Darmstadt, Germany) and 70 µL of internal standard working solution described in [36], and filtered using a methanol-preconditioned 96-well filter plate (Pall Corporation, Ann Arbor, MI, USA) loaded onto a vacuum manifold into an Armadillo high-performance 96-well PCR plate (Thermo Fisher Scientific). Samples were analyzed using a TriVersa NanoMate system (Advion, Ithaca, NY, USA) controlled by Chipsoft software (version 8.3.3, Advion). Data were acquired using Xcalibur software (version 3.0, Thermo Scientific, Waltham, MA, USA). Raw mass spectrometry data were analyzed using an in-house developed peak calling pipeline written in R programming language (source code available at https://github.com/UMCUGenetics/DIMS (accessed on 11 January 2015) that utilizes Human Metabolome DataBase (HMDB) for peak annotation with an accuracy of 5 ppm with respect to the theoretical m/z value, as described in detail in [36]. The web-based analysis tool MetaboAnalyst v.6.0 was used for statistical analysis (one factor) [61]. Metabolites with multiple possible annotations (isobaric compounds) were processed as single metabolite for statistical purposes.

### 4.11. Amino Acid and γ-Aminobutyric Acid (GABA) Analysis

Amino acid analysis was performed in 40 µL of zebrafish embryo (3 embryos/100 µL) extracts in 100% methanol (see Section 4.10 for preparation details) using an UPLC-MS/MS method described in [62]. GABA was quantified in 10 µL of zebrafish embryo (3 embryos/100 µL) or larvae (1 larva/100 µL) extracts in 100% methanol using an UPLC-MS/MS method described in [34].

### 4.12. Locomotion Analysis

ZebraBox system (ViewPoint Behavior Technology, Lyon, France) was used to track and quantify the locomotion of zebrafish embryos. Populations of 5 dpf old embryos or 10 dpf old larvae (1 embryo/well) were transferred to a 48-well flat-bottom plate (Greiner Bio-one CELLSTAR) containing 0.5 mL embryo E3 medium. Zebrafish embryos/larvae were allowed to acclimatize in the measurement chamber in the dark for 15 min prior to the measurement. Locomotion was assessed in the tracking mode using the following settings: background 15, inactivity threshold <1 mm/s, and burst activity threshold >30 mm/s. Temperature was maintained at 28 ± 1 °C. Locomotion was tracked in the dark without any intervention for 1 h. Movement trajectories were recorded and locomotion parameters were quantified with ZebraLab software (Viewpoint Behavior Technology, Lyon, France, https://www.viewpoint.fr/ (accessed on 23 May 2022).

### 4.13. Statistical Analysis

Data are presented as means ± SD. The number of zebrafish used for a specific experiment is indicated in the figure legends. Statistical analysis was performed using GraphPad Prism v.10 (GraphPad Software, San Diego, CA, USA). For comparison of two groups, Student’s *t*-test was used. For comparison of three or more groups, one-way ANOVA followed by Tukey’s post hoc test was used. The level of significance was set at *p* < 0.05.

## 5. Conclusions

In conclusion, we generated the first zebrafish model of HPP that shows multiple features of human disease and is suitable for studying the pathophysiology of HPP and testing novel treatments. We showed that Alpl has a function in vitamin B_6_ metabolism and bone mineralization in zebrafish. Untargeted metabolomics revealed a multitude of metabolic alterations occurring in response to Alpl deficiency, including but not limited to phosphoetanolamines, neurotransmitters, nucleotides, polyamines, and retinoids, suggesting potential interesting directions for follow-up research on the mechanisms of HPP in zebrafish. Furthermore, the normalization of multiple metabolic abnormalities in response to PN treatment suggests that vitamin B_6_ supplementation could be beneficial to HPP patients even in the absence of seizures. This study also revealed the limitations of performing metabolic research in zebrafish embryos, particularly related to the small embryo size that constrained the ability to analyze individual tissues/organs. Nevertheless, the data presented in this study clearly showed that this zebrafish model can serve as a valuable tool for investigating poorly understood aspects of TNSALP function and for developing improved therapies in the future.

## Figures and Tables

**Figure 1 ijms-26-03270-f001:**
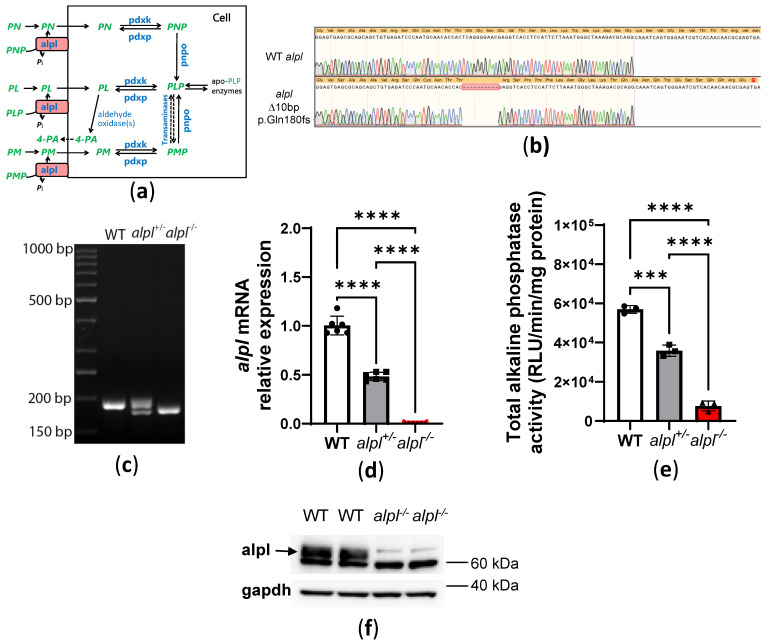
Generation and basic characterization of *alpl*^-/-^ zebrafish line. (**a**) Schematic representation of vitamin B_6_ metabolism. (**b**) Sanger sequencing results showing a 10 bp deletion at the CRISPR site in exon 5 of the *alpl* gene (c.623_632del, p.Gln180*fs*), which is predicted to result in a truncated protein. (**c**) Agarose electrophoresis of genotyping PCR products for the CRISPR site in exon 5 of the *alpl* gene in wild type (WT), *alp^+/-^, alpl^-/-^* 5 dpf old zebrafish embryos. (**d**) Relative *alpl* mRNA expression in WT, *alpl*^+/-^, and *alpl*^-/-^ 5 dpf old zebrafish embryos. Data are means from *n* = 3 pools (10 embryos per pool) per genotype measured in duplicate ± SD. **** *p* < 0.0001. (**e**) Total alkaline phosphatase activity in whole-embryo extracts of WT, *alpl*^+/-^, and *alpl*^-/-^ 5 dpf old zebrafish. RLU, relative light unit. Data are means from *n* = 3 pools (10 embryos per pool) per genotype ± SD. **** *p* < 0.0001 and *** *p* < 0.001. (**f**) Alpl protein expression in total WT and *alpl*^-/-^ 5 dpf zebrafish embryo extracts showing lack of alpl protein in *alpl*^-/-^ embryos (predicted molecular weight of alpl protein (NP_957301.2) is 62.5 kDa). Data are from pools of *n* = 32 embryos per genotype. Gapdh protein expression was used as the loading control.

**Figure 2 ijms-26-03270-f002:**
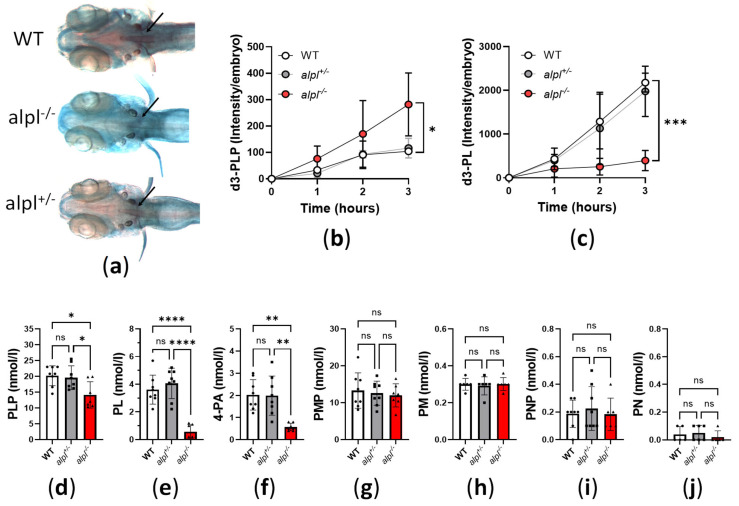
Impaired bone mineralization and abnormal vitamin B_6_ metabolism in *alpl* knockout zebrafish. (**a**) Decreased alizarin red staining of mineralized structures (notochord, indicated by arrows) in *alpl*^-/-^ compared to WT and *alpl*^+/-^ embryos at 5 dpf suggests a negative effect of alpl deficiency on bone mineralization. (**b**) Accumulation of d3-PLP and (**c**) decreased production of d3-pyridoxal (d3-PL) in *alpl*^-/-^ compared to WT and *alpl*^+/-^ embryos incubated with 100 µM d3-PLP in embryo water (E3) for 3 h at 28 °C. Data are means from *n* = 3 pools (3 embryos per pool) per time point and genotype ± SD. (**d**) Steady-state PLP, (**e**) pyridoxal (PL), (**f**) 4-pyridoxic acid (4-PA), (**g**) pyridoxamine 5′-phospate (PMP), (**h**) pyridoxamine (PM), (**i**) pyridoxine 5′-phosphate (PNP), and (**j**) pyridoxine (PN) concentrations in whole embryo extracts. Data are means from *n* = 8 pools (3 embryos per pool) per genotype ± SD. **** *p* < 0.0001, *** *p* < 0.001 ** *p* < 0.01, * *p* < 0.05, and ns—not significant (*p* > 0.05).

**Figure 3 ijms-26-03270-f003:**
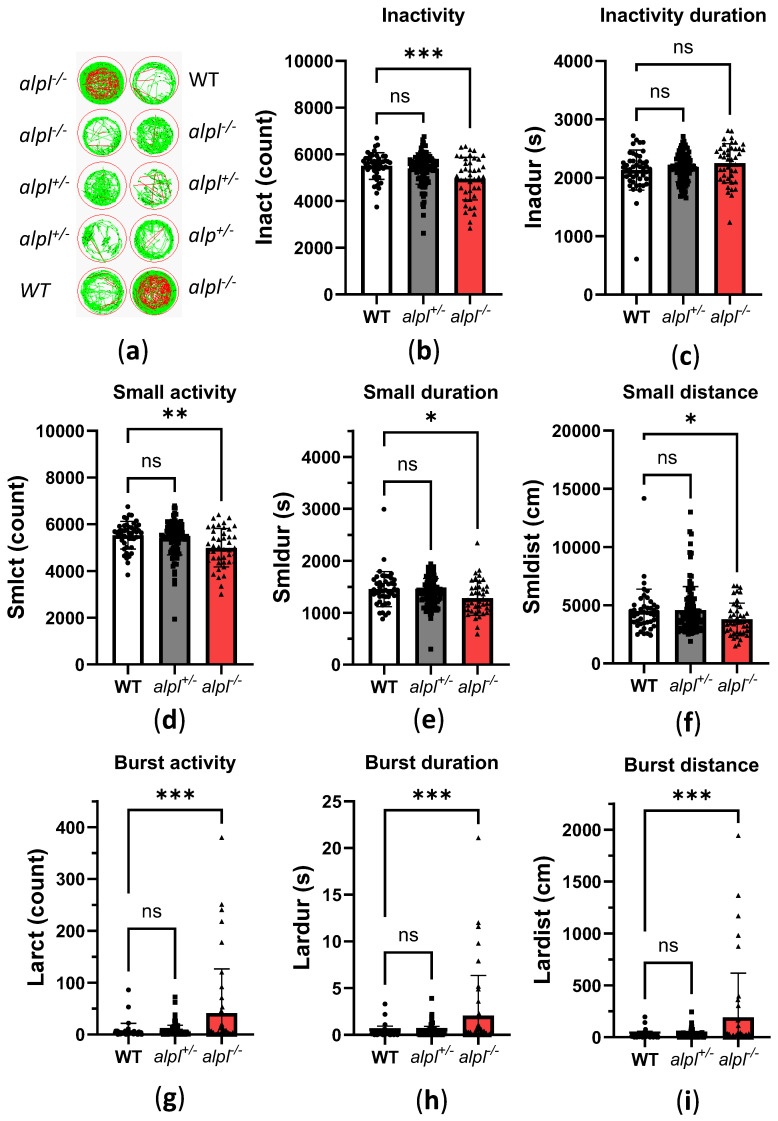
Locomotion analysis of 5 dpf old WT, *alpl*^+/-^, and *alpl*^-/-^ zebrafish embryos. (**a**) Examples of swimming trajectories of 5 dpf old embryos recorded with Zebrabox for 1 h at 28 °C in the dark. Green—movement speed < 30 mm/s (small activity), red—movement speed > 30 mm/s (burst activity), black—no movement (inactivity). (**b**) Inactivity count. (**c**) Duration of inactivity. (**d**) Small activity count. (**e**) Duration of small activity. (**f**) Distance swum during small activity. (**g**) Burst activity count. (**h**) Duration of burst activity. (**i**) Distance swum during burst activity. Data are means from *n* = 45 (WT), *n* = 105 (*alpl*^+/-^), and *n* = 42 (*alpl*^-/-^) embryos ± SD. *** *p* < 0.001, ** *p* < 0.01, * *p* < 0.05, and ns—not significant (*p* > 0.05).

**Figure 4 ijms-26-03270-f004:**
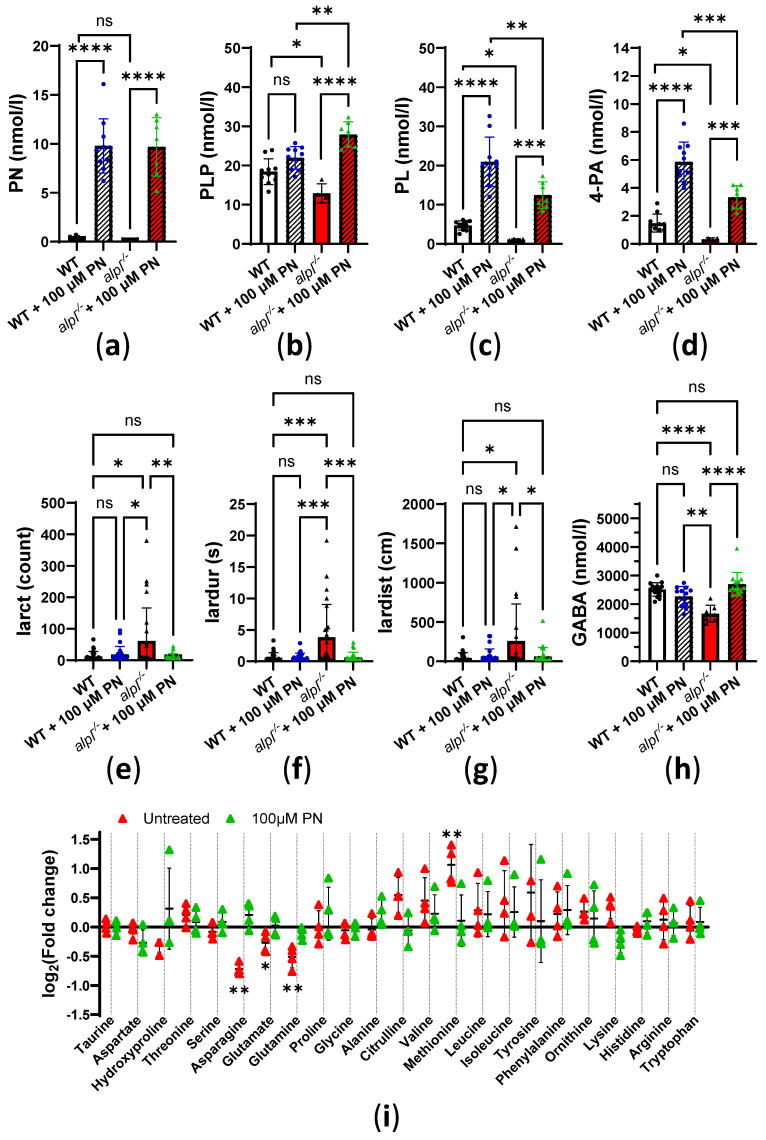
Normalization of biochemical and locomotion abnormalities in 5 dpf old *alpl*^-/-^ zebrafish embryos after 72 h continuous treatment with 100 µM pyridoxine (PN). Concentrations of (**a**) pyridoxine (PN), (**b**) PLP, (**c**) pyridoxal (PL), and (**d**) 4-pyridoxic acid (4-PA) measured in whole 5 dpf old embryo extracts. Data are means from *n* = 4–10 pools (3 embryos per pool) per genotype ± SD. Zebrabox analysis of (**e**) burst activity count, (**f**) duration of burst activity, and (**g**) distance swum during burst activity in 5 dpf old embryos measured for 1 h at 28 °C in the dark. Data are means from *n* = 23–25 embryos per genotype and treatment ± SD. (**h**) γ-Aminobutyric acid (GABA) concentrations. Data are means from *n* = 7–15 pools (3 embryos per pool) per genotype and treatment ± SD. (**i**) Fold-change analysis of amino acid concentrations in untreated *alpl*^-/-^ and 100 µM PN-treated *alpl*^-/-^ zebrafish embryos compared to the corresponding WT group. Data are from *n* = 4 pools (3 embryos per pool). **** *p* < 0.0001, *** *p* < 0.001, ** *p* < 0.01, * *p* < 0.05, and ns—not significant (*p* > 0.05).

**Figure 5 ijms-26-03270-f005:**
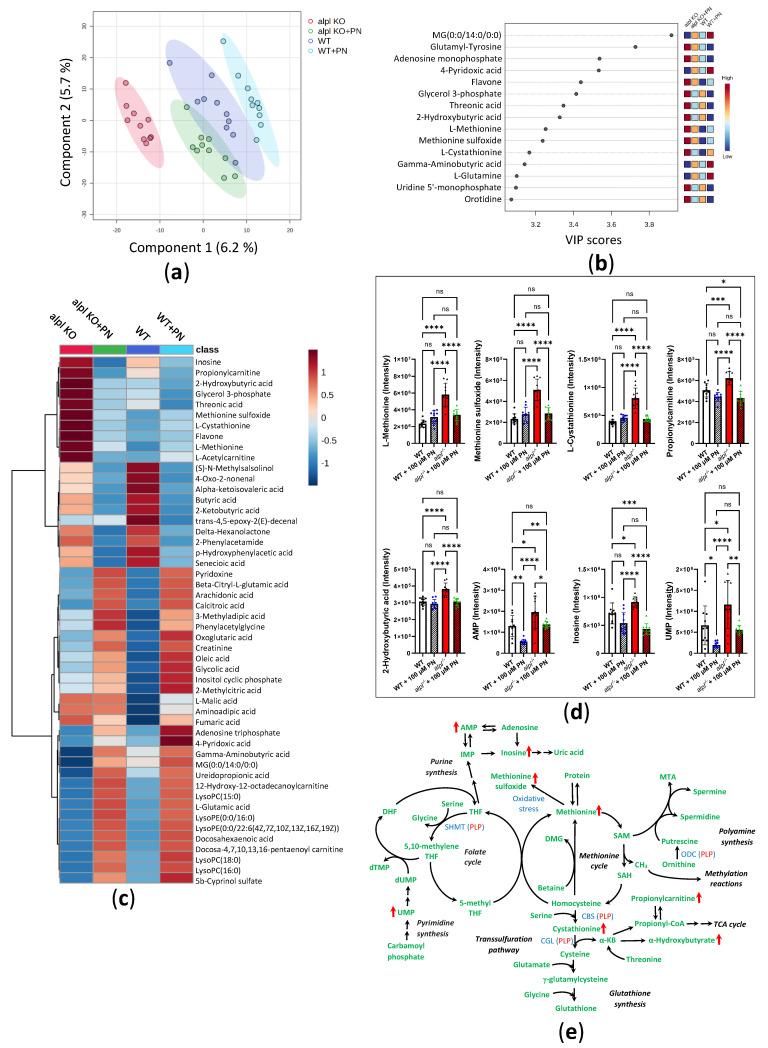
Global metabolic response to pyridoxine (PN) treatment in 5 dpf old WT and *alpl*^-/-^ zebrafish embryos. (**a**) Partial least squares-discriminant analysis (PLS-DA) scores plot (principal component 1 (*x*-axis) and component 2 (*Y*-axis)). The explained variances are shown in brackets. 95% confidence intervals are shown for each group. (**b**) Important metabolites identified by PLS-DA for principal component 1. Metabolites with the highest variable importance in projection (VIP) scores are shown. The colored boxes on the right indicate the relative intensity of the corresponding metabolite in each group. (**c**) Heatmap visualization of group-average intensities of the 50 highest-ranking metabolites based on one-way ANOVA results. Euclidean distance and Ward’s clustering algorithm were used for the hierarchical clustering of metabolites. (**d**) Intensities of a selection of the highest-ranking metabolites based on one-way ANOVA results. Data are means from *n* = 10 pools (3 embryos per pool) per genotype and treatment group ± SD. **** *p* < 0.0001, *** *p* < 0.001, ** *p* < 0.01, * *p* < 0.05, and ns—not significant (*p* > 0.05); comparisons as indicated in the graphs. (**e**) Schematic visualization of metabolic pathways where strongest effects of Alpl deficiency were observed followed by normalization after pyridoxine treatment. Metabolites are shown in green; red arrows indicate the effect of Alpl deficiency. Key PLP-dependent enzymes are shown in blue: SHMT, serine hydroxymethyltransferase (EC 2.1.2.1); CBS, cystathionine-β-synthase (EC 4.2.1.22); CGL, cystathionine γ-lyase (EC 4.4.1.1); ODC, ornithine decarboxylase (EC 4.1.1.17).

**Figure 6 ijms-26-03270-f006:**
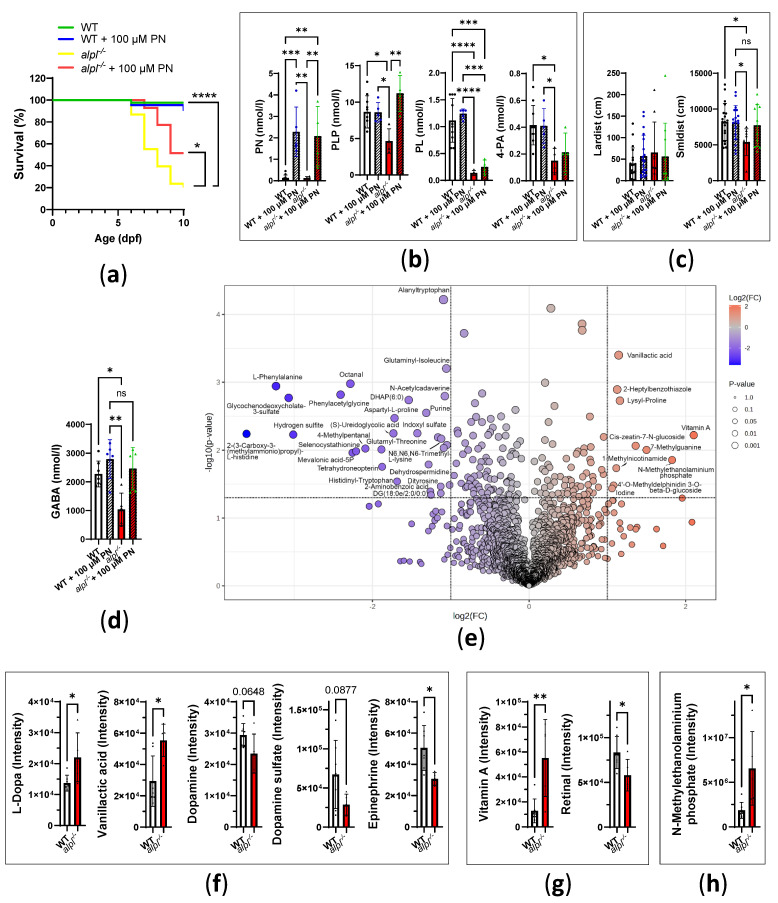
Progression of metabolic abnormalities in 10 dpf old *alpl*^-/-^ zebrafish larvae. (**a**) Kaplan–Meier survival curves of untreated and 100 µM PN-treated WT and *alpl*^-/-^ zebrafish until 10 dpf. Data are from *n* = 18–42 zebrafish per genotype and treatment condition. (**b**) Concentrations of pyridoxine (PN), PLP, pyridoxal (PL) and its degradation product 4-pyridoxic acid (4-PA) in untreated and 100 µM PN-treated WT and *alpl*^-/-^ larvae. Data are means from *n* = 4–10 larvae per genotype and treatment condition ± SD. (**c**) Distance swum during burst activity (lardist, movement speed > 30 mm/s) and during small activity (smldist, movement speed < 30 mm/s) during 1 h measurement using Zebrabox at 28 °C in the dark. Data are means from *n* = 9–19 larvae per genotype and treatment condition ± SD. (**d**) Concentrations of GABA in untreated and 100 µM PN-treated WT and *alpl*^-/-^ larvae. Data are means from *n* = 5–7 larvae per genotype and treatment condition ± SD. (**e**) Important features in metabolome of *alpl*^-/-^ larvae selected by volcano plot with fold change (FC) threshold equal to 2 (*x*-axis) and *t*-tests threshold of *p* < 0.05 (*y*-axis). (**f**) Accumulation of L-Dopa and vanillactic acid, and decreased level of dopamine, dopamine sulfate (dopamine 4-sulfate and dopamine 3-O-sulfate) and epinephrine in *alpl*^-/-^ larvae suggest impaired activity of aromatic L-amino acid decarboxylase. (**g**) Accumulation of vitamin A (retinol) and decreased level of retinal in *alpl*^-/-^ larvae. (**h**) Accumulation of N-methylethanolaminium phosphate in *alpl*^-/-^ larvae. Data are means from *n* = 5–7 larvae per genotype ± SD (panels E-H). **** *p* < 0.0001, *** *p* < 0.001, ** *p* < 0.01, and * *p* < 0.05.

## Data Availability

Data are available within this manuscript, figures, Appendix A. Raw DI-HRMS data are available upon request.

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
