# Peer review of "Broad Vitamin B_6_-Related Metabolic Disturbances in a Zebrafish Model of Hypophosphatasia (TNSALP-Deficiency)"

_ijms, 2025, doi:10.3390/ijms26073270_

Round 1

Reviewer 1 Report

Comments and Suggestions for Authors

Summary and general comments:

The manuscript describes the creation and characterization of a zebrafish model of hypophosphatasia due to TNSALP deficiency. This is of value to the literature as an additional animal model (after mice) of the disease, useful for drug screening and for understanding the underlying biochemical mechanisms of disease. In particular, the metabolomic data is useful for future work. The text is well-written and data well-presented. Overall, the data presented fully supports conclusions of the manuscript and I have no major concerns, provided my comments below are addressed.

Reviewer comments:

Generally, the gene is termed ALPL/alpl or another name depending on the species) and the protein quite consistently TNSALP. In the text, a deficiency of the protein is often termed TNSALP deficiency. However, in the title it is called ALPL-deficiency. I don’t think the zebrafish protein product has a name beyond ‘alkaline phosphatase’ but I think your data shows quite conclusively that it is homologous to ‘TNSALP’ in other vertebrates such as humans. I think, therefore, that it may be better to stick to TNSALP deficiency in the title.

I was particularly struck by the large accumulation of methionine and its sulfoxide. Although, as the authors mention, this has been observed in a mouse model of TNSALP deficiency, to my knowledge this has not been observed in other models of metabolic disorders causing B6 deficiency (an example is https://pubmed.ncbi.nlm.nih.gov/29061647/ where there is a decrease alongside ALDH7A1 deficiency). The authors mention in the discussion that because it is normalized by PN administration, methionine accumulation is likely due to the impaired activity of PLP-dependent enzymes, but this appears to contradict data from other disorders of B6 metabolism. Can the authors justify their statement further by, for example, providing the exact PLP-dependent enzymatic steps that could be affected?

Line 26: Add ‘the’ between ‘generated’ and ‘first’.

Figure 1A: Add aldehyde oxidase(s) to the step between PL and 4-PA.

Figure 2: Why is pyridoxine not shown? Probably, levels are low and unchanged, but it would be nice to show for completeness.

Line 155: tree should be ‘three’

Figure 4e-h: The most important finding of this data (as stated in the text) is that the movement phenotype in PN-treated alpl-/- zebrafish is not significantly different to WT fish, but there is no error bar showing this in the figure. Please add this.

Fig4i: It would be informative to see the individual data points in this figure, as in the bar charts above.

Line 363: ‘plays a role’

Line 396: Seizures ‘is’ should be seizures ‘are’.

Line 424: remove’ of’.

Author Response

Comment 1: Generally, the gene is termed ALPL/alpl or another name depending on the species) and the protein quite consistently TNSALP. In the text, a deficiency of the protein is often termed TNSALP deficiency. However, in the title it is called ALPL-deficiency. I don’t think the zebrafish protein product has a name beyond ‘alkaline phosphatase’ but I think your data shows quite conclusively that it is homologous to ‘TNSALP’ in other vertebrates such as humans. I think, therefore, that it may be better to stick to TNSALP deficiency in the title.

Response to Reviewer 1 comment 1:

We thank reviewer for pointing out this inconsistency. We changed ALPL to TNSALP in the title to keep it in line with the manuscript text.

Comment 2: I was particularly struck by the large accumulation of methionine and its sulfoxide. Although, as the authors mention, this has been observed in a mouse model of TNSALP deficiency, to my knowledge this has not been observed in other models of metabolic disorders causing B6 deficiency (an example is https://pubmed.ncbi.nlm.nih.gov/29061647/ where there is a decrease alongside ALDH7A1 deficiency). The authors mention in the discussion that because it is normalized by PN administration, methionine accumulation is likely due to the impaired activity of PLP-dependent enzymes, but this appears to contradict data from other disorders of B6 metabolism. Can the authors justify their statement further by, for example, providing the exact PLP-dependent enzymatic steps that could be affected?

Response to Reviewer 1 comment 2:

We thank reviewer for the comment. As discussed in the manuscript, metabolomics data in 5 dpf old alpl KO embryos consistently point towards a defect in transsulfuration pathway, which, in our opinion is the underlying cause of methionine and methionine sulfoxide accumulation in 5dpf alpl KO embryos. In humans,  defect in transsulfuration pathway, specifically caused by pathogenic variants in CBS gene coding for cystathionine beta-synthase, leads to accumulation of homocysteine and methionine, biomarkers of the disease. Our metabolomics data show accumulation of cystathionine, product of CBS and substrate for the next PLP-dependent enzyme in trassulfuration patway, cystathionine gamma-lyase (CGL) (Figure 5e), suggesting decreased activity of PLP-dependent CGL. In addition, there could possibly be complex contribution of various PLP-dependent transaminases, which is difficult to estimate in this setting.

The fact that different amino acid changes are observed in zebrafish model of ALDH7A1 deficiency compared to ALPL deficiency could be due to the distinct consequences that each specific deficiency has on amino acid metabolism in combination with PLP depletion. E.g. ALDH7A1 deficiency leads not only to reduction of PLP but also depletion and accumulation of various metabolites related to defect in lysine degradation pathway, which in itself could affect amino acid metabolism (in distinct manner compared to ALPL deficiency).

To clarify our reasoning we added specific enzymes that, in our opinion, are responsible for the observed effect (Lines 271-274):

These pathways contain vitamin B6-dependent enzymes (a.o. CBS and CGL, see Figure 5e), and the fact that all three metabolites were normalized after PN-treatment indicated that the effects were likely mediated by these enzymes.

Comment 3: Line 26: Add ‘the’ between ‘generated’ and ‘first’.

Response to Reviewer 1 comment 3:

Done.

Comment 4: Figure 1A: Add aldehyde oxidase(s) to the step between PL and 4-PA.

Response to Reviewer 1 comment 4:

We have adjuster Figure 1A according to the recommendation of the reviewer.

Comment 5: Figure 2: Why is pyridoxine not shown? Probably, levels are low and unchanged, but it would be nice to show for completeness.

Response to Reviewer 1 comment 5:

Indeed, reviewer is right, pyridoxine levels are extremely low and similar in WT and alpl KO zebrafish. However, to keep it complete we added pyridoxine data as panel (j) in Figure 2.

Comment 6: Line 155: tree should be ‘three’

Response to Reviewer 1 comment 6:

We corrected the typing mistake.

Comment 7: Figure 4e-h: The most important finding of this data (as stated in the text) is that the movement phenotype in PN-treated alpl-/- zebrafish is not significantly different to WT fish, but there is no error bar showing this in the figure. Please add this.

Response to Reviewer 1 comment 7:

We added the comparison of the groups mentioned by the reviewer in Figure 4e-h, they were indeed not significantly different.

Comment 8: Fig4i: It would be informative to see the individual data points in this figure, as in the bar charts above.

Response to Reviewer 1 comment 8:

We adjusted Figure 4i to include individual values, mean and SD.

Comment 9: Line 363: ‘plays a role’

Response to Reviewer 1 comment 9:

We corrected the mistake.

Comment 10: Line 396: Seizures ‘is’ should be seizures ‘are’.

Response to Reviewer 1 comment 10:

We corrected the mistake.

Comment 11: Line 424: remove’ of’.

Response to Reviewer 1 comment 11:

We corrected the mistake.

Reviewer 2 Report

Comments and Suggestions for Authors

Dear Editor, thank you for the opportunity to review the article entitled "Broad metabolic perturbations related to vitamin B6 in a zebrafish model of hypophosphatasia (ALPL deficiency)". The article is interesting and the techniques are in line with the study proposal and make the results found reliable. The article is good in all parts. My few suggestions and questions are for the authors' reflection. "

1- First, the order in which the introduction, results, etc. are presented.

I suggest that the text be presented in the following order: introduction, materials and methods, results, discussion, and conclusion. If this is the journal's rule, disregard it, but if not, please note that the order presented is confusing and different from that normally presented in original articles.

2- Introduction

The last paragraph, lines 76 to 87. It needs to be summarized and more direct.

3- Why was the GADPH protein chosen as the control?

4- Were the results found expected within the hypotheses planned before the study?

5- All the images and graphs are very good and congratulations to the authors for providing the original images. But would it be possible to have an illustrative figure showing the experimental designer of the study?

6- The techniques used to show the objective of the study are coherent. Could any other molecular or non-molecular technique also be used?

7- I suggest that the authors detail how to improve future directions.

8- I am aware that these study models do not include effect size, but it would be possible for the authors to present it in the analyses and comparisons. It is not mandatory, just a suggestion or question.

9- The article's conclusion topic must be included.

10- I missed a paragraph or sentence about the clinical benefits that the results bring in practical terms.

Overall, the article is good and interesting with chosen techniques that consolidate the results found in accordance with the proposed objective. My suggestions or questions were to contribute and for the authors' reflection.

Author Response

 Comment 1: First, the order in which the introduction, results, etc. are presented.

I suggest that the text be presented in the following order: introduction, materials and methods, results, discussion, and conclusion. If this is the journal's rule, disregard it, but if not, please note that the order presented is confusing and different from that normally presented in original articles.

Response to Reviewer 2 comment 1:

We thank reviewer for the suggestion. However, we followed the guidelines of the journal to structure the manuscript.

Comment 2: Introduction

The last paragraph, lines 76 to 87. It needs to be summarized and more direct.

Response to Reviewer 2 comment 2:

We thank reviewer for the suggestion. We shortened the last paragraph to include only essential information:

Zebrafish (Danio rerio) is a promising model organism to study human disease [27], including HPP [28,29]. Zebrafish have 4 genes coding for alkaline phosphatases: two for intestinal alkaline phosphatases alpi.1 and alpi.2 (gene duplication), one for alkaline phosphatase 3 alp3 (also expressed in intestine), and one for tissue-nonspecific alkaline phosphatase alpl, which also shows high degree of genetic conservation with human ALPL  [28,29]. In the present study we generated the first alpl-/- zebrafish line using CRISPR/Cas9 gene-editing technology. Biochemical and behavioral characterization of alpl-/- zebrafish showed that they display multiple features of infantile HPP., including de-creased bone mineralization, abnormal vitamin B6 metabolism, abnormalities in neuro-transmitter levels and pyridoxine-responsive seizures, as well as N-methylethanolaminium phosphate accumulation. Therefore, this new animal model could be used to gain insight in less understood aspects of HPP pathophysiology as well as for rapid screening of novel treatments.

Comment 3: Why was the GADPH protein chosen as the control?

Response to Reviewer 2 comment 3:

GAPDH (along with actin and tubulin) is a commonly used loading control in immunoblotting. We are aware that there is a discussion whether GAPDH is a suitable loading control, since it is involved in metabolism. In our particular case there was no indication that alpl knock-out could affect glucose metabolism, in which GAPDH is involved.

Comment 4: Were the results found expected within the hypotheses planned before the study?

Response to Reviewer 2 comment 4:

We showed that Alpl deficient zebrafish model that we created recapitulates several key features of human hypophosphatasia and is useful to gain insight in both basic function of Alpl as well as screening for novel therapies. Therefore, we think that the goal of our study is attained.

Comment 5: All the images and graphs are very good and congratulations to the authors for providing the original images. But would it be possible to have an illustrative figure showing the experimental designer of the study?

Response to Reviewer 2 comment 5:

We added a schematic overview of experimental setup as Supplemental Figure S6.

Figure S6 Overview of the experimental setup with 10 dpf old zebrafish larvae.

Comment 6: The techniques used to show the objective of the study are coherent. Could any other molecular or non-molecular technique also be used?

Response to Reviewer 2 comment 6:

In the present study we used untargeted metabolomics screening to map broad metabolic consequences of Alpl deficiency and the response to vitamin B6 (pyridoxine) treatment. Next step could be investigation of specific affected metabolic pathways using quantitative targeted techniques.

Comment 7- I suggest that the authors detail how to improve future directions.

Response to Reviewer 2 comment 7:

We thank the reviewer for the suggestion. We added the following text to address it (lines 455-461):

It must be noted that due to the contribution of isobaric compounds to the levels of AMP, inosine and adenosine (not changed in alpl-/- embryos, data not shown) determined with DI-HRMS, future follow up research using targeted methods is required to clarify the involvement of Alpl in the regulation of purine metabolism. This could contribute to better understanding of the mechanisms of chronic pain and help to develop new treatments to improve quality of life of HPP patients.

Comment 8: I am aware that these study models do not include effect size, but it would be possible for the authors to present it in the analyses and comparisons. It is not mandatory, just a suggestion or question.

Response to Reviewer 2 comment 8:

We thank the reviewer for the suggestion. Indeed, in this type of studies commonly only statistical significance of the differences are evaluated and reported, and we did so in our manuscript, using standard statistical methods. We doubt that calculating and including effect sizes in our study would affect or improve interpretation of the data.

Comment 9: The article's conclusion topic must be included.

Response to Reviewer 2 comment 9:

We thank the reviewer for pointing this out. We included a separate ‘5. Conclusions’ section (lines 664-679).

Comment 10: I missed a paragraph or sentence about the clinical benefits that the results bring in practical terms.

Response to Reviewer 2 comment 10:

To address the reviewers point we included the following sentence in the ‘Conclusions’ section (lines 672-674):

‘Furthermore, normalization of multiple metabolic abnormalities in response to PN treatment suggests that vitamin B6 supplementation could be beneficial to HPP patients even in the absence of seizures.’

Round 2

Reviewer 2 Report

Comments and Suggestions for Authors

Dear editor, after checking the changes and responses from the authors, I recommend accepting the manuscript.